# Finite-Element Analysis of High-Strength Steel Extended End-Plate Connections under Cyclic Loading

**DOI:** 10.3390/ma15082912

**Published:** 2022-04-15

**Authors:** Tao Lin, Zhan Wang, Fangxin Hu, Peng Wang

**Affiliations:** 1School of Civil Engineering and Transportation, South China University of Technology, Guangzhou 510640, China; 201410101220@mail.scut.edu.cn (T.L.); wangzhan@scut.edu.cn (Z.W.); 2State Key Laboratory of Subtropical Building Science, South China University of Technology, Guangzhou 510640, China; 3School of Environment and Civil Engineering, Dongguan University of Technology, Dongguan 523808, China; wangpeng@dgut.edu.cn

**Keywords:** high-strength steel, end-plate, connection, cyclic, finite-element analysis

## Abstract

In order to examine the seismic behavior of high-strength steel extended end-plate connections, a three-dimensional efficient finite-element model in Abaqus was established subjected to cyclic loading at the beam end. Geometrical dimensions, boundary conditions, element types, contact properties between the bolts, end-plate and column flange, and material cyclic constitutive models were described in detail. Geometry and material nonlinearity were adequately considered. In particular, a cyclic plasticity model for high-strength steels was employed that was easily calibrated based on the tension coupon test, so as to describe the complicated cyclic hardening and softening response. The simulated results of the finite-element model were compared to the test ones in terms of both deformation modes and hysteresis loops. The results showed that the bending deformation of the end-plate and column flange was accurately captured, and the gap phenomena among the bolt nuts, the end-plate, and column flange was described in a satisfactory manner as well. The hysteresis loops from the simulation agreed well with the test results, reproducing the pinched shape due to the end-plate gap evolution under cyclic loading as well as the quite plump shape with stable energy dissipation when the panel zone dominated the cyclic response. Therefore, the accuracy of the finite-element model was verified and it provided a strong benchmark tool for investigating the cyclic or seismic performance of this kind of connection. The connection failure including the bolt fracture and cracking in the end-plate needs further numerical study by calibrating accurate material failure models for high-strength steels and bolts.

## 1. Introduction

Extended end-plate moment connections have been widely used in steel frame structures due to their simplicity and easy constructability. In fact, in this connection, the joint is realized by fastening the beam to the column through an end-plate welded to the beam end and its production requires only shop-welding and field-bolting. Since this detail is relatively simple compared to other connection techniques, frames adopting this joint typology are fast to erect. Another positive feature regards the usage of an extended end-plate because, if the design process is properly carried out, this joint has a considerable elastic stiffness and a balanced energy dissipation mechanism is achieved between the bolts and the end-plate to maximize its energy dissipation capacity. In this way, the weld fracture problems that have been revealed in the past Northridge and Kobe earthquakes [1,2] in traditional welded joints can be avoided.

So far, there have been extensive studies on the seismic behavior of steel beam-to-column extended end-plate connections. Some representative experimental [3,4,5,6,7] and numerical [8,9,10,11,12,13] work of high quality on this type of connection can be found. Nowadays, high-strength structural steels with a nominal yield strength of at least 460 MPa have been developed and are increasingly recognized as structural materials to arrive at more economical or reliable solutions in structural design [14,15]. Since the material properties of high-strength steels are significantly different from those of conventional mild or low-alloy steels, in strength and ductility [16], their connection or structural behavior should deviate from each other and it is quite necessary to quantify the influence of the adoption of high-strength steels with a higher strength but lower ductility than conventional steels [17,18,19,20]. This is in more urgent need when the seismic performance is concerned, as the material ductility is relevant for seismic safety. Whether high-strength steel end-plate connections survive sufficient loading cycles under earthquakes up to a certain amplitude, for example, to a 4% story drift ratio in the AISC Seismic Provisions [21] for application in special moment frames with high ductility, deserves to be investigated.

There have been some reports on the structural behavior of high-strength steel end-plate connections. For example, Girão Coelho and Bijlaard [22,23,24] have conducted an experimental program consisting of both single-extended and flush end-plate connections. The end-plates were made up of the high-strength steels S460, S690, and S960 (nominal yield strength of 460 MPa, 690 MPa, 960 MPa, respectively), in Europe. The nonlinear behavior was characterized and the test results show that current design provisions can be extrapolated for stiffness and strength of those high-strength steel end-plate connections, which could also achieve reasonable rotation demands. Li et al. [25] carried out a numerical study on high-strength steel bolted end-plate beam-to-column joints by developing a finite element model to capture the effects of material and geometric nonlinearities. After verifying the accuracy in the prediction of the ultimate flexural resistances and moment-rotation curves, they continued to conduct a parametric study to investigate the effects of key design parameters on the behavior of bolted beam-to-column joints with double-extended high-strength steel end-plates. They finally found that the initial stiffness and plastic flexural resistance of the high-strength steel beam-to-column joints were overestimated by the current codes of practice. Jia et al. [26] performed parametric analysis on high-strength steel end-plate joint subjected to static loads. The influences of the column web and flange thickness, stiffening ribs, the steel grade and thickness of the end-plate, the bolt diameter, the compression ratio on the column, and the friction coefficient on the elastic stiffness, yield flexural resistance, and ductility were investigated. They found that the Eurocode 3 method for the flexural capacity was reasonable while the method for the initial stiffness showed poor accuracy. As for the seismic behavior of high-strength steel end-plate connections, Kim et al. [27] investigated experimentally the cyclic behavior of extended end-plate connections made up of SHN490 and SM490 steels and discovered that their cumulative energy dissipation was 60–70 kNm. Sun et al. [28] examined the hysteretic behavior of Q690 high-strength steel extended and flush end-plate connections, and concluded that the ultimate connection rotation capacity ranged from 0.02–0.04 rad. After comparison, they found that the ductility requirement in Eurocode 3 seemed to be unable to be directly used in the design of high-strength steel end-plate connections. They also developed a modified multilinear model to take into account the effects of the bolt bending deformation and the beam flange rotation and then calibrated a hysteretic model embedded in the software OpenSees and a cumulative damage model as well.

The mentioned research includes a limited number of experimental tests. In order to provide a comprehensive and in-depth understanding of the high-strength steel end-plate connection behavior, an accurate finite element model to be employed for parametric analysis is particularly important. However, there still lacks a satisfactory numerical modeling study, especially in terms of the cyclic behavior of high-strength steel end-plate connections under earthquake loading. The complicated contact behavior involved in those connections makes it difficult to achieve accurate predictions. The distinct material response under monotonic and cyclic loadings contributes to this challenge, although some numerical modeling studies for those connections under monotonic loading have been well conducted. Therefore, this paper is devoted to building a finite-element model to predict accurately the hysteretic behavior of high-strength steel extended end-plate connections. Such a model will provide a powerful tool for carrying out further analysis.

## 2. Finite-Element Model

The general-purpose finite analysis software Abaqus [29] was adopted to develop an efficient and accurate numerical model. Static analysis was conducted using its standard module. Both material and geometrical nonlinearities were incorporated. The model details are described as follows.

### 2.1. Geometrical Dimensions

Typical beam and column dimensions were designed based on the common practice in low- and mid-rise steel frame structures. The benchmark end-plate connection assembly is shown in Figure 1, which consists of an H-shaped beam H320 × 160 × 8 × 14 (mm) and an H-shaped column H200 × 160 × 12 × 16 (mm). To confirm the “strong column-weak beam” requirement, the beam was supposed to be made of Q355 steel (nominal yield strength of 355 MPa), while the column was Q690 steel (nominal yield strength of 690 MPa). Continuity plates, whose steel grade and thickness are identical to those of the beam flange, are aligned to the beam flanges so as to introduce the beam moment into the column in an effective way. The extended end-plate was also supposed to be made of Q690 steel and its thickness was 16 mm in the benchmark model, so as to exploit the more economic design benefit while investigating the effect of steel grade. A total of ten class 10.9 M20 high-strength bolts, arranged in five rows and two columns, serve to connect the end-plate and the column flange. The end-plate has the same width as the column and beam, and a total height of 500 mm. Bolt holes in the end-plate were symmetrically prepared to meet the requirements of a minimum bolt spacing of 3 *d*_0_, a minimum edge distance of 1.5 *d*_0_, and a minimum end distance of 2 *d*_0_, where *d*_0_ is the bolt hole diameter (22 mm), as per the Chinese Specification [30]. The welded connection between the beam and the end-plate, as well as that between the continuity plates and the column, were not explicitly modeled as they were just tied together. This should impose little influence on the connection behavior, since those groove or fillet welds should be strong enough, if a proper design procedure is followed, in common practice.

The total height of the column was determined as 3000 mm, which represents a typical story height. The distance between the beam end and the column centerline was chosen to be 3000 mm as well, obtained as half of a typical span of 6000 mm, if the beam end is considered as an inflection point when the frame structure is subjected to lateral load.

### 2.2. Boundary Conditions

To simulate the gravity load on the column, an initial axial compression, which is 30% of the nominal axial yield resistance of the column section, was imposed on the top of the column. Since both column ends were taken as the inflection points in the frame structure, they should be hinged in the plane of the connection so as to release the rotational degree-of-freedom. Because the axial compression was present, the bottom column end was pin-supported, while the top one was roller-supported. Thus, the axial deformation of the column was released as well.

Cyclic displacement-based loading was imposed at the beam end in the plane of the beam-to-column connection according to the prescribed protocol in the AISC Seismic Provisions [21], which is shown in Figure 2. The story drift angle, as the controlled parameter, refers to the ratio of the displacement at the beam end over the distance to the column centerline (i.e., 3000 mm). In addition, the out-of-plane displacement and rotation at the beam ends were restricted. This is in accordance with the out-of-plane bracing in practice.

### 2.3. Element Types and Meshes

The reduced-integration 8-node linear brick element of type C3D8R in Abaqus was used for meshing the beam, column, end-plate, and high-strength bolts. This element is popular in modeling since it is computationally efficient and can effectively avoid shear locking, but this element may cause hourglass-mode problems. In this regard, three layers of elements were established across the plate thickness direction. Previous studies have evidenced the validity of this meshing strategy.

The structured meshing technique was assigned to arrive at an appropriate geometrically symmetric meshing appearance, especially for the high-strength bolts and the end-plate. A much finer mesh in the vicinity of the bolted end-plate connection was incorporated, where the mesh size is about 8 mm. In this way, the computational effort could be reduced while the highly nonlinear end-plate connection behavior is accurately captured.

### 2.4. Contact Modeling

Modeling of the bolts in the end-plate connection was simplified as symmetrical cylinders, as shown in Figure 3. The washer was neglected; while the bolt head and nut were modeled as identical, and their height was taken as the average, 20 mm, of their real dimensions, including the thickness of the washer. The diameter of the bolt shank was taken as the nominal diameter of 20 mm, which means the screw threads were overlooked. In this way, a radical clearance of 1 mm between the bolt shank and the wall of the corresponding bolt hole exists, since the bolt hole diameter (see Figure 1) is 2 mm larger than the bolt shank. The height of the bolt shank was simply determined as the total thickness of connected plates. Previous studies on conventional steel end-plate connections have proved the effectiveness of this simplified modeling [10].

There are four pairs of contact interaction in this bolted end-plate connection: the contact between the end-plate and the column flange, the contact between the bolt nuts and the end-plate, the contact between the bolt nuts and the column flange, and lastly, the contact between the bolt shank and the walls of bolt holes in the end-plate and column flange. The locations of those contact pairs are shown in Figure 3. In terms of modeling in Abaqus, the contact behavior is described by the “Surface-to-Surface Contact” defined in the “Interaction” module. Before the establishment of a surface-to-surface contact pair, the contact properties should be defined, including those for the tangent contact and normal contact directions. Coulomb friction is used for the former and the friction coefficient is selected as 0.45 for slip-critical high-strength bolted connections; while hard contact is used for the latter, which is used to avoid the penetration between the elements in the contact pair. In this way, the contact simulation is as close to the actual situations as possible.

The “Bolt Load” command in Abaqus is used to apply pretension at the middle cross-section of bolts, as shown in Figure 3. According to the Chinese Specification [30], the pretension force is 195 kN. The bolts in the end-plate connection sustain both shear force and axial force. Due to the highly nonlinear behavior resulting from the complicated contact interaction, the bolts, the end-plate, and the column flange should be carefully meshed for rapid and acceptable convergence in computation.

### 2.5. Material Modeling

#### 2.5.1. Material Modeling of Steel Plates

Since the seismic behavior is investigated in this paper, the connection assembly is subjected to cyclic loadings. In this regard, an accurate constitutive model for cyclic plasticity of structural steels is required, rather than a simple monotonic stress–strain model in which an isotropic hardening is usually assumed. Traditional cyclic plasticity models for structural steels include simple isotropic hardening, bilinear or nonlinear kinematic hardening, and mixed or combined hardening models. For example, a combined hardening model was implemented in Abaqus for metal plasticity, which was firstly proposed by Chaboche [31,32]. This model consists of multiple nonlinear kinematic hardening components and an isotropic hardening component. However, it is rather difficult to calibrate the Chaboche model when the monotonic stress–strain curve from tension coupon tests is the only available material data. Recently, one of the authors has proposed specific cyclic plasticity models for conventional and high-strength structural steels and established a calibration procedure based on the full-range stress–strain curve [16,33,34]. These models and their associated calibration method have been verified with respect to Q235 (nominal yield strength of 235 MPa), Q355, Q550 (nominal yield strength of 550 MPa), and Q690 steels in China. In fact, both cyclic hardening and softening phenomena have been considered in those models by a sophisticated combination of both isotropic and kinematic hardening components, which contributes to a better description of cyclic stress–strain behavior. According to the method proposed therein, the parameters of Q355 and Q690 steels can be calibrated in a straightforward way, without any iteration or trial and error. The model parameters are shown in Table 1, based on the tension coupon tests on steel plates used for subsequent experimental studies. *E* is the elastic modulus, *f**_y_* is the yield stress, $\varepsilon_{\mathrm{st}}^{p}$ is the plastic strain at the end of the yield plateau, and ${\bar{\varepsilon }}_{\mathrm{st}}^{p}$ is the threshold plastic strain to distinguish the plateau and hardening regions for those steels with a distinct yield plateau; $Q_{i}^{s}$, $b_{i}^{s}$ and $Q_{i}^{l}$, $b_{i}^{l}$ are short-term and long-term isotropic hardening parameters, respectively; $C_{i}^{s}$, $\gamma_{i}^{s}$ and $C_{i}^{l}$, $\gamma_{i}^{l}$ are short-term and long-term kinematic hardening parameters, respectively; $c^{s}$ and $c^{l}$ are short-term and long-term memory scalers, respectively. In addition, the Poisson’s ratio was taken as 0.3 for steels.

#### 2.5.2. Material Modeling of High-Strength Bolts

The traditional bilinear kinematic hardening model was adopted for the high-strength bolts, as shown in Figure 4. The plastic hardening modulus was determined based on the yield stress and the ultimate stress and strain. The stress and strain parameters, which were true values derived from the nominal stress and strain, were also determined according to the tension tests. The Poisson’s ratio was taken as 0.3.

## 3. Verification

### 3.1. Description of Test Program

A group of five bolted extended end-plate connection specimens was prepared and tested. The first specimen, labeled as S1, is the benchmark one with dimensions identical to those shown in Figure 1. The second and third specimens are identical to the benchmark one except for the thickness of the end-plate. The second specimen, S2, consists of a 12 mm-thick end-plate; while the end-plate thickness is 8 mm in the third specimen, S3. The fourth specimen, S4, was intended to examine the influence of panel zone strength, so the column section was taken as H200 × 160 × 8 × 16 (mm) where a thinner (8 mm-thick) column web than the benchmark was adopted. The fifth specimen, S5, was used to investigate the effect of column flange thickness, so the column section was taken as H200 × 160 × 12 × 12 (mm) where a thinner (12 mm-thick) column flange than the benchmark was used. For the last two specimens, the other design parameters remained the same as the benchmark.

Cyclic loading tests were carried out in a test setup as shown in Figure 5, in accordance with the same boundary conditions as that shown in Figure 1. Both column ends were connected to pin supports. An actuator was employed to apply the cyclic load at the beam end, while a jack on the top pin support was used to apply the initial axial compression on the column. The distance between the loading point at the beam end and the column centerline is 3000 mm, while that between the pins at the column top and bottom is 3000 mm as well. Lateral bracing was arranged close to the beam end, so as to restrict the potential out-of-plane buckling. An embedded load cell in the actuator measured the instantaneous force, while a displacement transducer was set to measure the beam end displacement for load control per Figure 2.

### 3.2. Comparison of Tests and Finite-Element Analysis

#### 3.2.1. Deformation Modes

For the extended end-plate connections, as the imposed displacement amplitudes increased, the gap between the end-plate and column flange in the tensile zone generally increased, as well as the residual bending deformation of the end-plate. Finally, the specimens failed due either to bolt fracture or the cracking and subsequent fracture of the joint region between the end-plate and beam flanges. Experimental results showed that the specimens S1, S2, S4, and S4 exhibited the first bolt fracture at the first negative peak of 6%, the first positive peak of 6%, the first negative peak of 8%, and the second negative peak of 5% loading stages, respectively; while for the specimen S3, the end-plate started cracking at the 3% loading stage and the eventual fracture occurred at the first positive peak of 5%, due to the relatively thin end-plate adopted. Since the current finite-element model cannot capture the fracture phenomenon, the simulated deformation modes were compared with the experimental ones just at the last peak amplitude before failure triggered by the bolt or end-plate fracture, as shown in Figure 6, Figure 7, Figure 8, Figure 9 and Figure 10 for each specimen, where von Mises stress contours are included as well. The bending deformation of the end-plate was well simulated, as well as the opening gap between the end-plate and the column flange. The shear deformation of the panel zone in specimen S4 was obviously captured, as shown in Figure 9 because a weak column panel was designed for this specimen. The column flange bending in the tension zone, as shown in Figure 10 for specimen S5, was also notable and described with satisfactory accuracy, in the case of weak column flange design.

#### 3.2.2. Hysteresis Loops

Hysteresis loops between the moment and the story drift angle are shown in Figure 11 for each specimen. The story drift angle, which has been explained previously, is the ratio of the measured displacement at the beam end over the distance to the column centerline (i.e., 3000 mm), while the moment refers to that at the column face which is calculated as the product of the measured force at the beam end and the distance to the face of column flange (i.e., 3000 mm minus half of the column cross-section height). Except for specimen S4, an obvious pinching phenomenon was noted during the unloading stage in the hysteresis loops of the rest specimens. This is due to the gap evolution between the end-plate and the column flange. At the loading stage, the bolt nuts were in tight contact with the end-plate and column flange; while at the unloading stage, the gap developed, which indicated the separation between the bolt nuts and plate surfaces, due to the plastic strain accumulation of those components subjected to cyclic loading. This made the force or moment in the hysteresis loops nearly unchanged with the increase in the displacement at the beam end. Based on the comparison, the finite-element simulation was able to well simulate this characteristic. It is also notable that in the last several cycles, there was considerable deviation between the experimental and simulated curves, which was mainly caused by the inability of the finite-element model to describe the bolt fracture and end-plate cracking phenomena.

The hysteresis loop in specimen S4 was relatively plump since its characteristic was dominated by the weak panel zone subjected to cyclic shear. The satisfactory hysteresis behavior was accurately captured by the simulation until the bolt fracture occurred eventually.

Above all, the established finite-element model was able to provide a quite accurate prediction for the elastoplastic hysteresis behavior of high-strength steel bolted end-plate connections under cyclic loading.

## 4. Conclusions

The three-dimensional finite-element model of high-strength steel extended end-plate connection in frame structures was established. The detailed modeling techniques were illustrated and the model was then verified by typical experimental tests. The following conclusions can be drawn:(1)The proposed finite-element model could give a quite accurate prediction for the pinched hysteresis loops and deformation modes of bolted end-plate connections. It proved the rationality of selected element types, contact interaction properties, and, in particular, the material constitutive model for high-strength structural steels. This finite-element model provided a powerful tool for further studying the connection performance under earthquakes.(2)The weak panel zone design demonstrated its stable and plump hysteresis behavior, and the panel zone had a much larger energy dissipation capacity than the bolted end-plate, in the case of them all made up of high-strength steels. It indicated the need to further investigate the potential to take advantage of the superior seismic performance of panel zones in connection design. The weak column flange design, however, did not show much impact on the overall hysteresis behavior of the end-plate connection.(3)Because the bolt fracture or cracking in the end-plate was not taken into account in modeling, the connection failures in experimental tests were not simulated. It is suggested to calibrate accurate material failure models for high-strength steels and bolts so that they can be implemented in Abaqus to capture the connection failure. This is required for reasonable identification of the ultimate resistance, ductility, and energy dissipation capacity of high-strength steel end-plate connections.(4)Generally, high-strength steel extended end-plate connections survived cycles up to a story drift angle amplitude of 5%; while when the panel zone dominated the plastic response, such an amplitude was as high as 7%.

## Figures and Tables

**Figure 1 materials-15-02912-f001:**
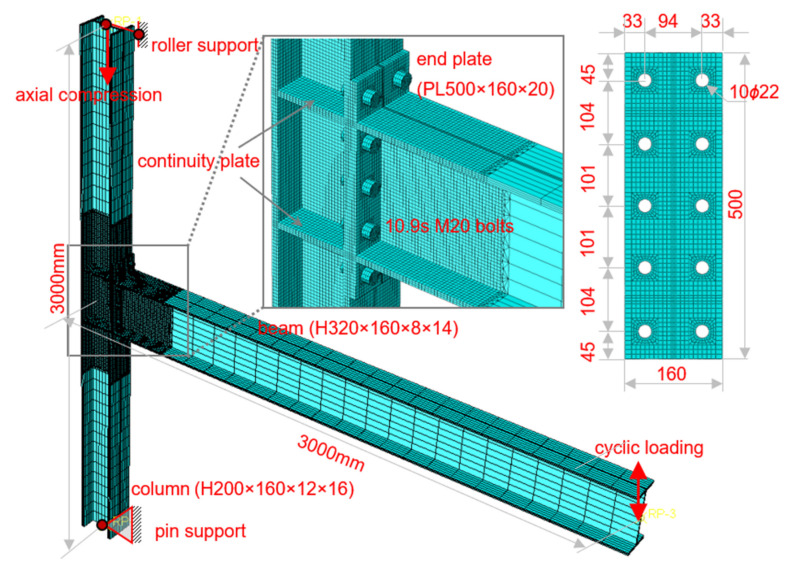
Benchmark finite-element model.

**Figure 2 materials-15-02912-f002:**
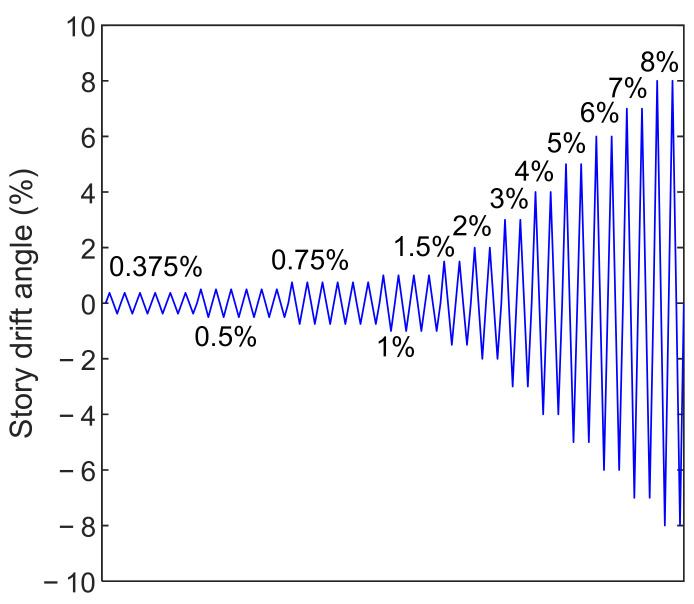
Cyclic loading protocol.

**Figure 3 materials-15-02912-f003:**
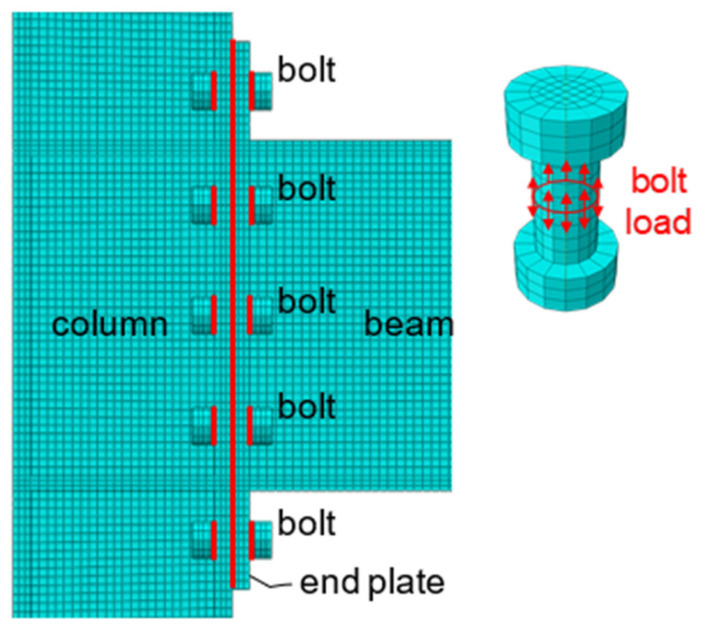
Contact pairs and bolt load.

**Figure 4 materials-15-02912-f004:**
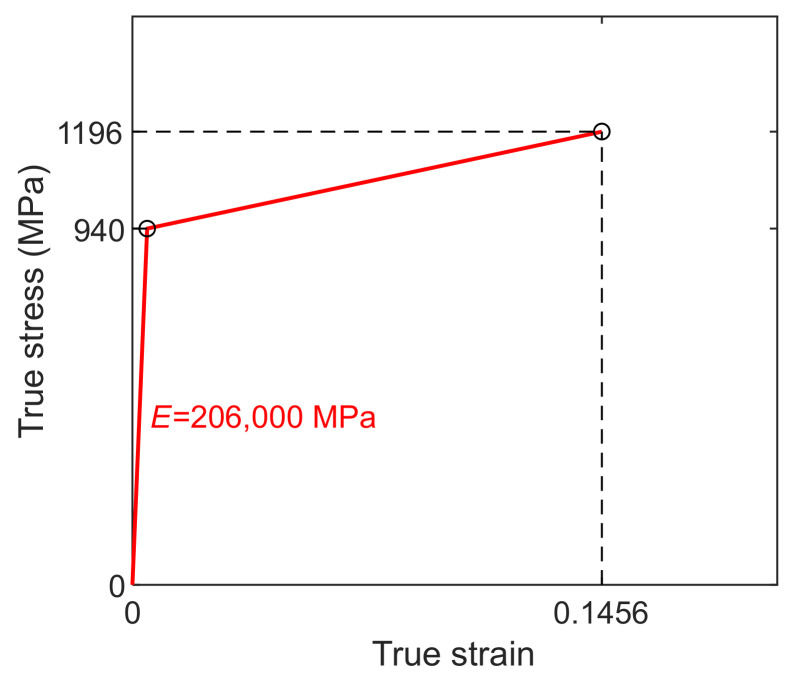
Stress–strain model of high-strength bolts.

**Figure 5 materials-15-02912-f005:**
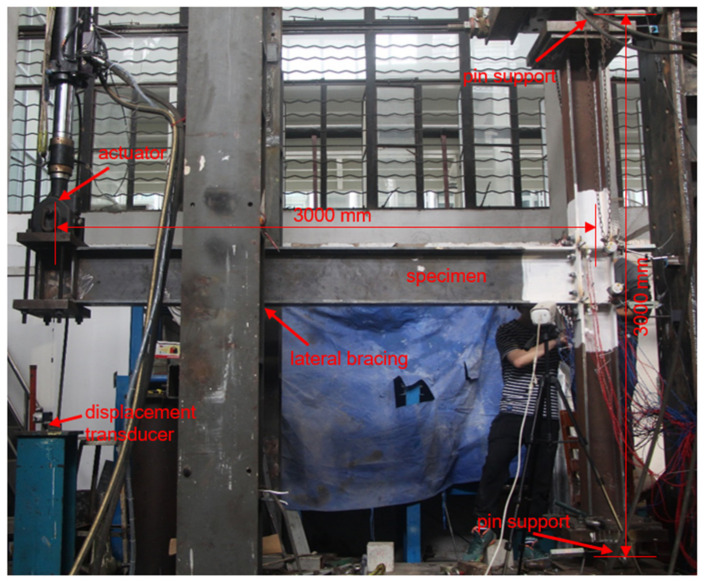
Test setup.

**Figure 6 materials-15-02912-f006:**
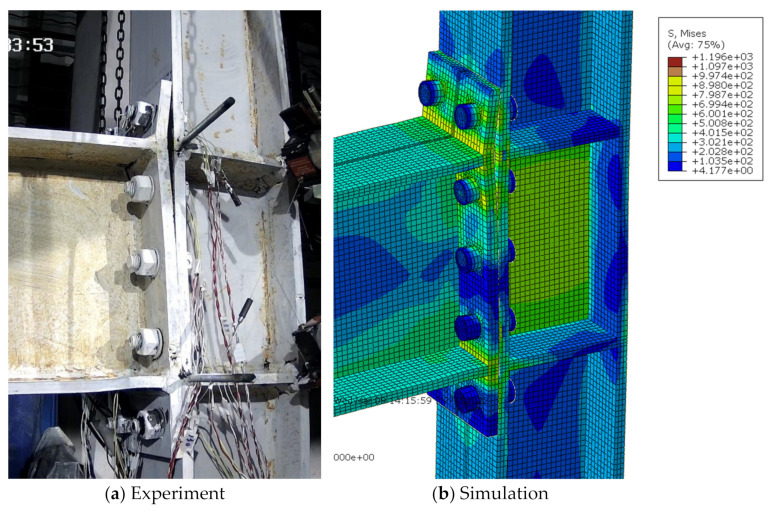
Comparison of experimental and simulated deformation modes in specimen S1 at the first positive peak of 6% loading stage.

**Figure 7 materials-15-02912-f007:**
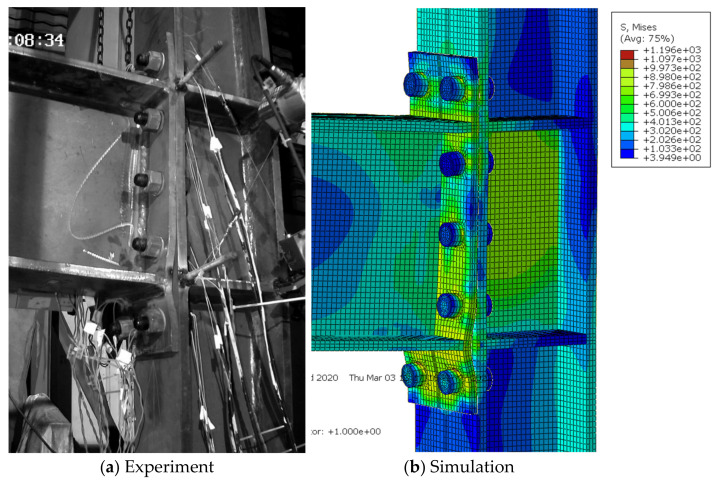
Comparison of experimental and simulated deformation modes in specimen S2 at the second negative peak of 5% loading stage.

**Figure 8 materials-15-02912-f008:**
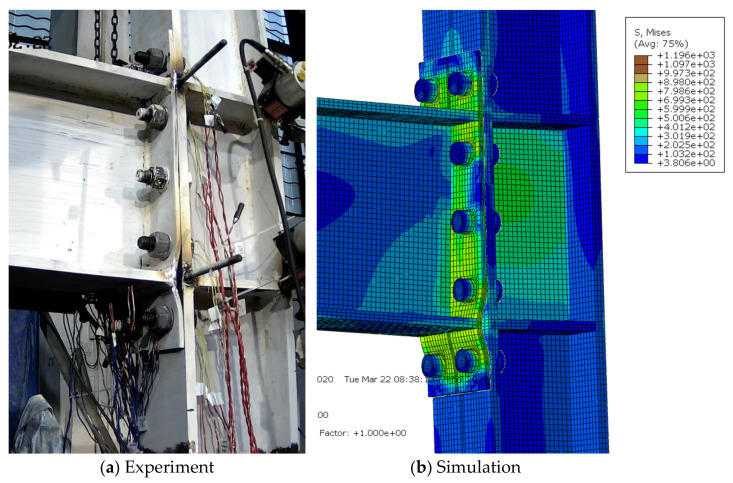
Comparison of experimental and simulated deformation modes in specimen S3 at the second negative peak of 4% loading stage.

**Figure 9 materials-15-02912-f009:**
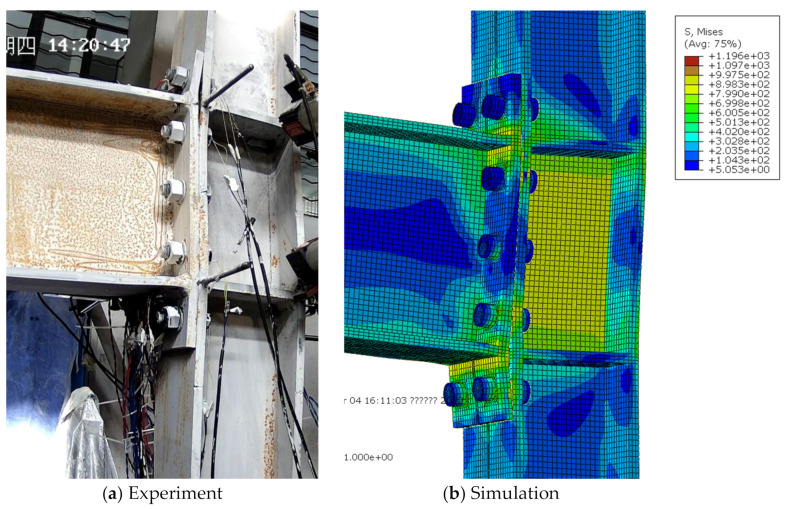
Comparison of experimental and simulated deformation modes in specimen S4 at the first positive peak of 8% loading stage.

**Figure 10 materials-15-02912-f010:**
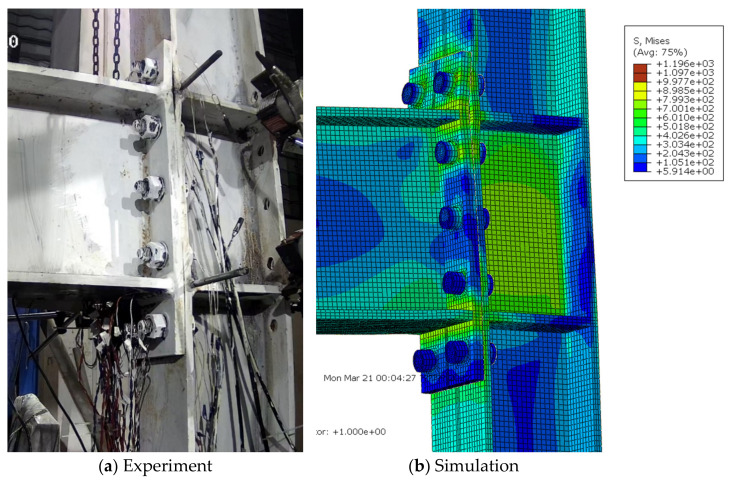
Comparison of experimental and simulated deformation modes in specimen S5 at the second positive peak of 5% loading stage.

**Figure 11 materials-15-02912-f011:**
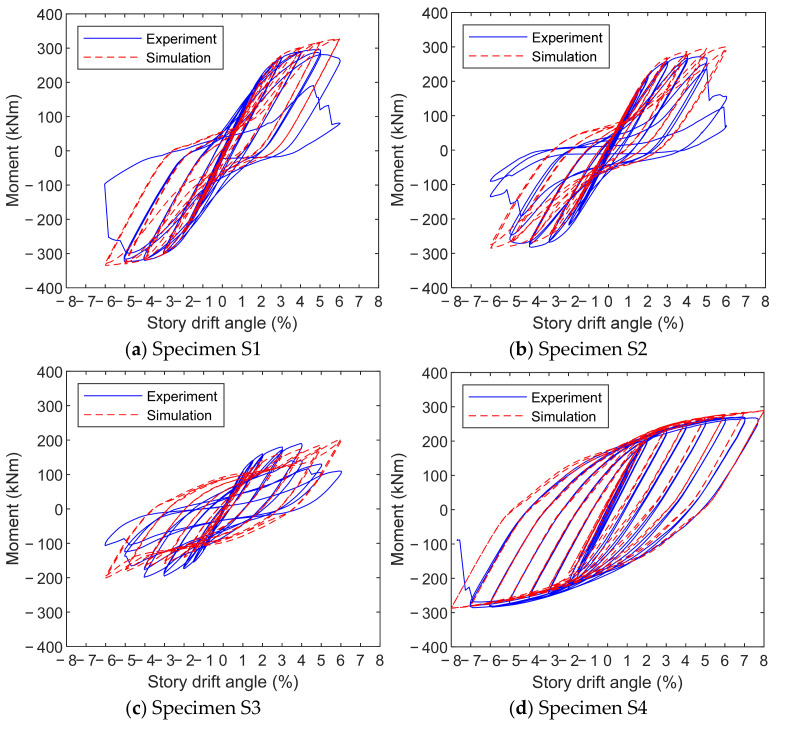
Comparison of experimental and simulated hysteresis loops.

**Table 1 materials-15-02912-t001:** Calibrated parameters of steel plates.

Steel Gr.	*E* (MPa)	$f_{y}$	$Q_{1}^{s}$	$Q_{2}^{s}$	$Q_{1}^{l}$	$Q_{2}^{l}$	$C_{1}^{s}$	$C_{2}^{s}$	$C_{1}^{l}$	$C_{2}^{l}$	$C_{3}^{l}$	$c^{s}$
	$\varepsilon_{\mathrm{st}}^{p}$	${\bar{\varepsilon }}_{\mathrm{st}}^{p}$	$b_{1}^{s}$	$b_{2}^{s}$	$b_{1}^{l}$	$b_{2}^{l}$	$\gamma_{1}^{s}$	$\gamma_{2}^{s}$	$\gamma_{1}^{l}$	$\gamma_{2}^{l}$	$\gamma_{3}^{l}$	$c^{l}$
Q355 (8 mm)	205,500	406	−203	0	108	0	203,000	40,600	3127	374	0	0.5
	1.2%	0.5%	300	0	25	0	3000	300	52	0	0	0.3
Q355 (14 mm)	203,500	368	−184	0	129	0	184,000	36,800	3560	377	0	0.5
	1.8%	0.5%	300	0	25	0	3000	300	52	0	0	0.3
Q690 (8 mm)	208,300	610	−122	−122	98	49	366,000	36,600	3672	34,678	272	0
	0	0	3000	300	35	650	3000	300	45	850	0	0.2
Q690 (12 mm)	205,800	775	−155	−155	30	15	465,000	46,500	928	8764	260	0
	0	0	3000	300	35	650	3000	300	45	850	0	0.2
Q690 (16 mm)	219,800	811	−162	−162	25	13	486,600	48,660	733	6919	266	0
	0	0	3000	300	35	650	3000	300	45	850	0	0.2

## Data Availability

The data presented in this study are available on request from the corresponding author. The data are not publicly available due to privacy.

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
