# Peer review of "Finite-Element Analysis of High-Strength Steel Extended End-Plate Connections under Cyclic Loading"

_materials, 2022, doi:10.3390/ma15082912_

Round 1

Reviewer 1 Report

The work describes a study about the fatigue analysis of a widely employed connection of steel frame structures. The numerical and the experimental procedures were accurately described, and the outcomes of the work are well reported.

The following points should be considered for the revised version of the manuscript:

  • Line 8, “model to taken into account the effects” change to “model to be taken into account the effects”.
  • Line 108, the description of the utilization of the “continuity plates” is not fully clear. Try to better clarify the usage of these plates, a figure could be helpful.
  • The particular in Figure 1 shows a fast transition of the mesh size from the fine to the coarse mesh. Usually, this kind of fast transition determines numerical errors, and a more gradual change is recommended. Did the author check this aspect together with the mesh convergence of the model?
  • Line 133, the expression “inflection points” is usually referred to the points where the deflection curve of the beam changes the sign of the curvature. The expression “restrain points” sounds more appropriate to refer to points where some constraints are applied.
  • Figure 3, the red lines indicating the contact pairs are shifted downwards.
  • Line 159, was any radial clearance between the bolt shank and the walls of bolt holes considered?
  • Fig.6 to Fig. 10, the contour legend must be added to each plot to improve their physical meaning. What kind of contour plot is reported? A displacement component or the total displacement?
  • It would be interesting to report some figures with the Von Mises stress or the plastic deformation to verify that the locations with higher stress or plastic strain correspond with those of experimental failure.
  • The authors reported a few references about the techniques of FE simulation for bolted joints, more references should be reported such as: https://doi.org/10.1016/j.compstruct.2020.113199; https://doi.org/10.1016/j.engstruct.2021.113368

Reviewer 2 Report

Construction materials made of steel are tested under both static and dynamic load. Very often, various types of structures are subjected to cyclic and fatigue loads. Determining the load-displacement characteristics is very important from the point of view of describing the behavior of the structural material for different frequency and load cycles. The presented comparative numerical and laboratory tests on histeresis loop five specimens are a valuable part of the article and allow for a better understanding of the momentum and story drift angle characteristics. Below are some comments and suggestions:

  1. In the introduction, it should be written the number of load cycles for the calculation of high strength steel end-plate connections under dynamic loading, (it should be added two literature items).
  2. In the article, numbering should be added to the individual chapters, it will significantly improve the readability of the article.
  3. In the chapter on „Finite-element model”, it should be written what stress-strength characteristics were used in the modeling for cyclic loads. In the subsection called „Material modeling” this has been partially explained, however, the information on the number of modeled cycles is interesting.
  4. For table 1, it should be written what the letters mean: f, Q, C, ε, b, γ, c.
  5. Lines 45, 57, 77, 79, 108, 112, 116, 118, 126, 128, 142, 161, 163, 184, 204, Table 1, Figure 4, it should be used a space between the digit and the unit.
  6. In the chapter on „Verification”, it should be written on the scale of the laboratory tests and what sensors and apparatus were used to measure the load and displacement.
  7. In Figures 6a, 7a, 8a, 9a, 10a, the scale relating to colors is missing.
  8. For the subsection on „Hysteresis loops” (line 286), the equations by which the hysteresis loop is drawn should be provided.
  9. In the last Chapter one conclusion should be written relating to the numerical values ​​obtained in the numerical and laboratory tests.

Round 2

Reviewer 1 Report

The authors discussed and solved the points I raised.

In my opinion, the paper can now be accepted for publication.